# Design and Research of Superimposed Force Sensor

**DOI:** 10.3390/mi16091069

**Published:** 2025-09-22

**Authors:** Genshang Wu, Jinggan Shao, Yicun Xu, Zhanshu He, Shifei Liu

**Affiliations:** 1Henan College of Transportation, Zhengzhou 450006, China; 18737136876@139.com (G.W.); sjgjygs@163.com (J.S.); 2Henan Jiaoyuan Engineering Technology Group Co., Ltd., Zhengzhou 450008, China; 3School of Mechanical and Power Engineering, Zhengzhou University, Zhengzhou 450001, China; xuyicun@zzu.edu.cn (Y.X.); hezhanshu@zzu.edu.cn (Z.H.)

**Keywords:** force standard machine, SolidWorks, finite element analysis, force sensor

## Abstract

The measurement accuracy and equipment stability of superposition-type force sensors are primarily influenced by the layout and number of individual force sensors. Analyzing this impact effect through experimental testing for each configuration would consume significant manpower, material resources, and financial costs. To efficiently analyze the influence of the number of paralleled individual sensors and their layout within a superposition-type force measurement instrument on overall device stability and force measurement accuracy, this paper employs SolidWorks to establish models of force instruments based on common superposition schemes. Subsequently, ANSYS is utilized to perform finite element analysis on models of different schemes, obtaining corresponding data on total deformation, stress, and simulated force values. The analysis results indicate that a relatively sparse sensor layout with symmetric arrangement around the center point of the base plate enhances overall stability, and the force measurement error can be controlled within several ten-thousandths. Furthermore, the more stable and higher-accuracy schemes identified through simulation analysis were compared with practical experimental results to analyze theoretical versus actual errors. The test results showed that when the three single force sensors are placed in a “Pin font” shape, the sum of the forces measured by each individual sensor differs from the sum of the forces measured by the superimposed sensors by only a few ten-thousandths, which is within the acceptable range.

## 1. Introduction

Force standard machines [1] are devices capable of generating standard force values for the verification and calibration of dynamometers or load cells, complying with national metrological regulations. With the continuous advancement of technology, force measurement and control are trending towards larger scales while demanding increasingly higher accuracy. Common force measurement instruments currently include superposition-type, deadweight-type, hydraulic-type, and lever-type instruments, among others [2,3,4]. Within the force verification system, compared to deadweight, lever, and hydraulic methods, superposition-type force standard machines (SFSMs) offer advantages such as a large force measurement range, high work efficiency, and a small footprint [5,6]. However, this method employs relative comparison measurement rather than absolute measurement. The force value uncertainty primarily depends on the performance indicators of the standard dynamometer, the series connection method of the dynamometer under test, the assembly quality of the standard machine, and the performance of the loading structure [7]. Therefore, research on superposition-type force measurement instruments is essential [8,9,10].

Improving the measurement performance of superposition-type force sensors and reducing measurement uncertainty has remained a persistent challenge. Hefei Jianghang Aircraft Equipment Co., Ltd. (Hefei, China) [11] analyzed and researched the hydraulic system control methods for SFSMs. To enhance measurement accuracy, they proposed three key control points for the system, reducing the verification cost of the force standard machine, achieving high-precision control of the superposition hydraulic system, and improving verification efficiency and data security. The 20 MN force standard machine project developed by Haidi Shan, Weiming Zhao, and Quanhong Liu from the Henan Provincial Institute of Metrology was successfully completed in 2017 [12]. This project partially overcame the shortcomings of SFSMs, enabling the verification and calibration of standard dynamometers and load cells, thereby improving force value stability and detection accuracy. Xiurong Wang [13] analyzed the structural configuration, advantages, and disadvantages of a 60 MN superposition-type force standard device. Subsequently, finite element analysis (FEA) was used to analyze the stress state of the device’s frame structure under maximum load conditions. Based on the simulation results, structural optimization was performed on areas with insufficient strength, enhancing the overall system’s stability and load-bearing capacity after optimization. Anyi Huang et al. [14] proposed a force transfer system with a load-equalizing structure for the 500 kN superposition section of a deadweight-superposition composite force standard machine. This system ensures force is evenly distributed to each sub-sensor, reduces azimuth error, mitigates the impact of rotational effects, and enhances metrological performance.

Current research on improving the stability and measurement accuracy of superposition-type force standard devices primarily focuses on optimizing factors such as system control and structural characteristics. These improvements have shown certain limitations in effectiveness and have not increased the maximum measurement range. Compared with the work of other researchers, this study can fundamentally solve the problems of limited measurement accuracy and poor equipment stability of the superimposed force standard device. This paper explores the impact of different schemes on the overall system stability and measurement accuracy by changing the layout and superposition number of single force sensors and finds the more reasonable one from multiple research schemes. Moreover, based on the model of the single force sensor and the number of parallel connections, a force standard machine with a larger capacity range can also be designed [15,16]. The force values involved in this design process reach tens of meganewtons (MN). However, analyzing the effects of different configurations experimentally one by one would consume substantial manpower, material resources, and financial costs [17]. Therefore, models of different schemes can be established using SolidWorks (2019 Version), and their effectiveness can be validated using ANSYS (2022 R1 Version) finite element analysis (FEA) software. Simulation data can then be compared to identify superior configurations, followed by experimental verification of the chosen scheme’s feasibility. This research not only significantly shortens the equipment development cycle but also enables a direct understanding of the stress concentration, structural deformation state and measurement force value error of the superimposed force standard device during operation. At the same time, it provides a theoretical basis for optimizing the structure and improving the measurement accuracy of the existing superimposed force standard device.

## 2. Scheme Design and Simulation Analysis

### 2.1. Design of Scheme

The primary structure of a superposition-type force standard device consists of a set of high-accuracy standard force transducers serving as the reference standard. A lower platen acts as the mounting fixture for these standard force transducers, while an upper platen serves as the load-bearing platform. The standard force transducers are distributed between the upper and lower platens. As shown in Figure 1, the device is a microcomputer-controlled superimposed force standard machine of Hangzhou Xin High-Tech Co., LTD. (Hangzhou, China). The left part of Figure 1 is the microcomputer control system, and the right part is the superimposed standard force gauge. The microcomputer control system can ensure the high precision of force value measurement and the high stability of equipment operation of this device. When using the device in Figure 1 to verify the force value, the force transducer under test is placed on the upper platen. A load is then applied hydraulically or mechanically, enabling comparative measurement to determine the metrological characteristics of the transducer under test. However, the sensors are arranged in parallel between the upper and lower platen; both the number of sensors and their layout significantly impact the overall system’s range, stability, and measurement accuracy. Therefore, it is crucial to maintain the number of sensors within a reasonable range. An excessive or insufficient number of sensors can increase design complexity and difficulties [18,19]. Simultaneously, the sensor layout must be rational: sensors should be neither excessively clustered nor overly dispersed. An arrangement that is as symmetrical and uniform as possible is essential to fully leverage the sensors’ performance and ensure system stability.

Based on an investigation of commercially available superposition-type force standard devices, the number of superimposed sensors typically ranges from 3 to 6, depending on the force range and equipment control structure. The layout methods for different numbers of sensors also vary. For example: 3 sensors: Linear arrangement, compact placement, placement with a certain separation distance. 4 sensors: “3 + 1” distribution, square distribution, annular distribution with a certain separation distance. 5 sensors: Annular distribution, “4 + 1” distribution. Since the design complexity and achievable force range for configurations using 6 sensors significantly exceed the requirements of this study, such configurations will not be considered. Following analysis and design, common layouts are illustrated in Figure 2, Figure 3 and Figure 4. In these figures, the sensors are simplified as circles for illustrative clarity.

### 2.2. Three-Dimensional Model Construction

The primary structure of the superposition-type force measurement model consists of an upper platen, a lower platen, and the sensors. The upper and lower platens are square plates measuring 400 × 400 mm with a specified thickness. According to different layout schemes of the sensors, appropriate threaded holes should be drilled at the pressure-bearing plate to accommodate the sensors placed in the middle. This can make the sensors more firmly fixed. To enhance simulation fidelity and ensure the results closely reflect real-world conditions, corresponding 3D models were created in SolidWorks at a 1:1 scale, based on the actual height and diameter of the sensors. The measurement process of the sensor height and the diameter of the indenter is shown in Figure 5 and Figure 6, where the diameter of the indenter of the sensor is the same as the diameter of the threaded hole at the pressure-bearing plate.

During the modeling process, a 0.2 mm groove was added to the contact surface between the upper side of the lower platen and the sensor, and the sensor model can be inlaid in the groove of the lower pressure-bearing plate. This recess serves two purposes: it defines the sensor placement position and increases the contact area. This design helps prevent positional shifts in the sensor caused by lateral force components during operation, thereby enhancing the overall stability of the device. Sensor models were adjusted in terms of quantity and position according to the different design schemes. One of these models is exemplified in Figure 7.

### 2.3. Finite Element Simulation Preprocessing

This simulation is designed to serve real-world experiments. Therefore, the conditions applied to the superimposed force standard machine during the finite element simulation process should all be various external forces given during the experiment. To clearly understand the various conditions required for the simulation, this section analyzes the working principle of the superimposed force value testing machine and provides various force and boundary conditions for simulation pretreatment based on the actual situation. The superposition-type force testing machine is shown in Figure 8. The upper section features four force measurement transducers, symmetrically positioned front and back. These transducers apply the required force to the parallel-connected force standard machine. The central section constitutes the main subject of this design (the superposition-type force standard machine). The lower section is a platform designed to support the parallel-connected force standard machine. This platform must be capable of withstanding the applied forces.

Pressure is applied through the four sensors rather than directly across the entire upper surface. This inevitably leads to localized stress concentration at certain points. Therefore, to ensure the upper pressure-bearing plate does not deform or even fracture due to excessive localized stress, theoretical analysis must be performed prior to experimentation [20]. Additionally, the deformation of the upper pressure-bearing plate must be considered. This is because during testing or in practical applications, a certain amount of deformation could impact the measurement uncertainty, introducing unpredictable errors that compromise data accuracy and degrade measurement precision [21]. Within the central section of the superposition-type force standard machine lie the sensors. These sensors are secured to the lower platen via screws and simultaneously bear the pressure applied to the upper pressure-bearing plate. This analysis focuses on the magnitude of the vertical force acting on the sensors. Since the sensors contact the upper pressure-bearing plate via their top surfaces, this interface will also experience some stress and deformation. All of these factors (stresses, deformations, forces) can be determined within the Finite Element Analysis model.

### 2.4. Material Properties and Boundary Conditions

The upper and lower pressure-bearing plates of the superimposed force sensor will be subjected to a considerable force during operation, to ensure that the pressure-bearing plate has sufficient strength during the force application process, the material of the pressure-bearing plate is set as structural steel in the simulation process. The main materials of the force sensor used in this test are carbon steel, stainless steel, and aluminum alloy, etc. Since most of the materials of the force sensor are steel, its material can also be set as structural steel in the simulation process, and its main attribute parameters are shown in Table 1.

The force applied to the upper surface of the upper pressure-bearing plate is not directly applied to the entire surface. Instead, it is applied at the top surfaces of the four force measurement transducers. Consequently, four distinct contact areas must be defined on the upper platen corresponding to these loading points. To avoid errors caused by unbalanced force application and even system collapse due to imbalance, and to ensure the safety of the experiment and the accuracy of the experimental data, the four positions for applying force should be at equal distances and symmetrically distributed from the center and edge of the upper pressure-bearing plate. The center positions of the four sides of the upper pressure-bearing plate can be marked to divide the upper pressure-bearing plate into four identical areas. A diagonal can be taken for each area, and the intersection point of the diagonals is a position for applying force, as shown in Figure 9.

Since the lower pressure-bearing plate needs to be in direct contact with the support plate of the testing machine and it is full face-to-face contact, there will be no deformation due to the force transmitted by the sensor above. Therefore, in this case, it can be considered that the lower end face of the lower pressure-bearing plate is directly fixed and then receives the force applied from above. For the addition of external force, here we select the situation where there are four sensors in the force standard machine. Each force application point on the upper end face is independently operated, and a force of 10,000 N is applied, with the direction set perpendicular to the upper end face and pointing towards the internal sensors.

By now, all the boundary conditions have been clarified: The force applied by each force value measurement sensor above should be the number of sensors in the superimposed force standard machine multiplied by the full scale and then divided by four, with the direction perpendicular to the upper pressure-bearing plate and downward. The lower pressure-bearing plate should be directly fixed. When conducting ANSYS simulation, settings should be made directly in the model operation to ensure that finite element analysis of the superimposed sensor can be carried out under the predetermined conditions.

## 3. Results

### 3.1. Total Deformation

The different scheme models of the above design were solved under the same mesh division, boundary conditions and solution operations, respectively, and the deformation amounts of different scheme models under the same simulation conditions were obtained. The simulation result of one scheme is shown in Figure 10.

The purpose of this finite element analysis is to analyze the superimposed force standard machines of different schemes, identify the changes produced by each model under the same boundary conditions, compare the impacts of different arrangements of the same number of sensors on the overall structure, and find the scheme with the minimum deformation. As shown in Figure 10, under the applied force of 10,000 N × the number of sensors, the upper pressure-bearing plate of the superimposed force sensor undergoes relatively obvious deformation, mainly concentrated in the four corners of the upper pressure-bearing plate, and the deformation gradually decreases towards the center of the upper pressure-bearing plate. The minimum deformation occurs at the center of the chassis. Since the chassis is stationary, the arrangement of the sensors will not affect it. Therefore, only the maximum deformation amount and its occurrence location are summarized, as shown in Table 2.

By analyzing Table 2, Under the application force of 10,000 N × the number of sensors, the maximum deformation of the upper pressure-bearing plate of different scheme models is approximately distributed between 0.4 mm and 0.6 mm, and the location where the maximum deformation occurs is all at the top Angle of the upper pressure-bearing plate. Moreover, different layout methods of sensors under the same number have a significant impact on the deformation amount. Therefore, when considering the total deformation amount of the superimposed force standard machine, it is necessary to avoid closely arranging several sensors together as much as possible. Instead, on the entire available chassis, the required sensors should be spaced at a certain distance and symmetrically arranged according to the center point of the chassis so as to minimize the maximum deformation of the superimposed force standard machine to the greatest extent. Among the above-mentioned schemes, the one with the smallest maximum deformation is when the three sensors are placed in a “Pin font” shape, with a deformation of 0.42848 mm. This scheme can enhance the stability of the equipment and reduce the measurement error caused by the deformation of the pressure-bearing plate.

### 3.2. Y-Direction Stress

The simulation of stress mainly involves analyzing the stress magnitudes at the upper end face of the cover plate and the contact surfaces of the four force value measurement sensor pressure heads, as well as at the lower end face of the cover plate and the contact surfaces of several force measurement sensor pressure heads. When the yield strength of the structural steel is known, it is determined which positions have stress higher than the yield strength. The simulation results are shown in Figure 11, Figure 11a shows that the stress at the cover plate is mainly concentrated at the contact position between the cover plate and the sensor, and the stress distribution is roughly the same as the shape of the sensor’s indenter. Figure 11b shows that the stress at the sensor is mainly concentrated at the indenter plane, while the rest of the stress is relatively evenly distributed.

In order to compare the differences in the post-processing results of parallel force standard machines with different combinations under the same boundary conditions, so that the optimal solution can be selected in actual manufacturing and theory can guide practice. After processing the simulation results of different schemes, the maximum stress, minimum stress and their occurrence locations are shown in Table 3. Among them, when the three sensors are placed in a “straight line”, due to structural imbalance, errors will occur during ansys simulation because the calculation results cannot be obtained, and they should be discarded.

As can be seen from Figure 11 and Table 3, the contact surface between the lower end face of the cover plate and the upper end face of the sensor is the place with the maximum stress value, while the rest of the areas have a relatively smooth stress distribution. When analyzing whether the yield strength of the structural steel is met, only whether the stress value of the contact surface is less than 250 MPa is considered. When considering the maximum stress of the parallel force standard machine, that is, ensuring the yield strength of the structural steel, it is necessary to avoid closely arranging several sensors together as much as possible. Instead, on the entire available chassis, the required sensors should be spaced at a certain distance and symmetrically arranged according to the center point of the chassis. To obtain the optimal parallel force standard machine, the maximum stress is within a safe range.

### 3.3. Force Value

To analyze the error between the total force values received by each force sensor and the total applied force values, in this case, the forces acting on each part within the complex model can be analyzed separately. Since a single force sensor will receive force values in the *X*-axis, *Y*-axis, and *Z*-axis directions, a vector arrow will be led out on the contact surface in the model to indicate the direction of the force. The vector arrows are composed of the component forces in the *X*-axis, *Y*-axis, and *Z*-axis directions, respectively. The force expression of a single sensor is shown in Figure 12. When analyzing the force value error, only the force value in the Y-direction needs to be analyzed.

The force in the sensor is the resultant force in the *X*-axis, *Y*-axis and *Z*-axis directions. The total sum of the component forces of each sensor in the Y-direction is statistically analyzed to determine whether it is equal to the total force values applied to the model by the four force value measurement sensors above, and the corresponding errors are calculated. The simulation data of the force values in the Y-direction for different schemes and the calculated error values are shown in Table 4, Table 5 and Table 6.

From the data in the above table, it can be seen that the total force applied by the force sensor in the Y direction has an error of one ten-thousandth to one ten-thousandth compared with the total force values applied by the four force value measurement sensors above to the model. The measurement of the external force by the sensor part in the entire model is basically consistent, and the error can be ignored. Moreover, this error will not increase with the increase in the number of parallel force sensors. Under the same number of force sensors, a relatively sparse arrangement can reduce the error. Among all the schemes, the error is the smallest when arranged in a “Pin font” shape.

## 4. Discussion

From the results of the simulation analysis, it can be known that in the designed scheme, the three sensors are placed in a “triangular” layout, making the entire equipment more stable and reducing the measurement error. This scheme can be selected to analyze what impacts it will have on the superimposed force sensor during the test process. To investigate whether the overlay force measurement model experiences deformation-induced errors under higher force loads and to determine if the sum of force measurements from individual sensors equals the externally applied force on the entire device, the following test was conducted. Three different sensor models (Manufacturer: high-precision HBM sensors produced by HBM Company of Germany, Accuracy Class 0.1, to facilitate the distinction of the force values measured by each sensor, they are numbered as 31715918, 52043494, and 52043493 based on the production information of the sensors) were connected in parallel for error analysis. As the force increased from 2 kN to 10 kN, the errors of the three different sensor models fluctuated around the order of thousandths, having a minor impact on the test results. These three sensor models were arranged in a triangle formation. A 100 kN force standard machine (Serial Number: 413, Accuracy Class 0.03) served as the force application device. The overall test setup is shown in Figure 13.

During this test, the 100 kN force standard machine applied force to the upper compression plate of the test device. The readings from the three force sensors were then measured. The actual combined reading was calculated based on the measured readings from these three sensors. This actual combined reading was then compared with the theoretical combined value of the three sensors obtained from ANSYS simulation to analyze the error between the experimental results and the theoretical model. The force applied by the 100 kN force standard machine in this test setup was incrementally increased according to specific values for separate experimental analysis [22]. The readings from the three sensors, the actual combined reading, and the error magnitude were recorded at different applied force levels. The final data results obtained are presented in Table 7.

Through simulation summary, it is concluded that when the equipment is placed in a “Pin font” layout, the stability of the equipment can be improved and the measurement error can be reduced, with the error being controlled within 0.01%. However, during the test process, due to factors such as the measurement environment, measurement personnel, and the characteristics of the test equipment, the measurement uncertainty will increase. Therefore, there is a certain deviation between the test and the simulation. As can be seen from the data in Table 7, the error between the actual parallel values and the theoretical parallel values of each sensor is controlled within a few ten-thousandths, and it will not change significantly with the increase or decrease in the force standard machine reading. All are within an acceptable range. It can be considered that the sensors are the same when placed separately or in parallel. It does not affect the accuracy when conducting experiments.

## 5. Conclusions

Based on the actual dimensions of the sensors, a three-dimensional model was constructed. Finite element analysis was then employed to evaluate the total deformation, stress and force values under different arrangements of three to five sensors. Experimental validation was subsequently conducted to quantify the error between theoretical predictions and actual measurements, yielding the following conclusions:(1)In the superimposed force measurement model, the layout of sensors should be spaced at certain intervals, and the entire chassis should be utilized as much as possible to reduce the deformation of the pressure-bearing plate under force, improve the stability of the overall equipment, and reduce the error caused by deformation when measuring the force value. Among them, when the three sensors are placed in a “Pin font” shape, the total deformation of the pressure-bearing plate is the smallest, which is 0.42848 mm. The position is located at the top corner of the upper pressure-bearing plate, while the center position of the lower chassis is the minimum deformation position of the entire model.(2)The stress of the superimposed force measurement model is mainly concentrated at the contact surface between the lower end face of the upper pressure-bearing plate and the upper end face of the sensor. Among them, when the three sensors are placed in a “Pin font” shape, the stress is the smallest, with a magnitude of 20.358 MPa, which is less than the yield strength of structural steel and meets the safety requirements.(3)The error between the total force value of each superimposed sensor in the Y direction and the applied force value is controlled within one ten-thousandth to one ten-thousandth. The error does not change significantly with the number and layout of sensors and remains within an acceptable range.(4)When the three sensors are arranged in a “Pin font” shape for the test, the sum of the forces measured by each sensor differs from the theoretical parallel value by only a few ten-thousandths of the indicated value of the force standard machine. This error will not fluctuate greatly with the increase in the test force value, meeting the corresponding requirements.

## Figures and Tables

**Figure 1 micromachines-16-01069-f001:**
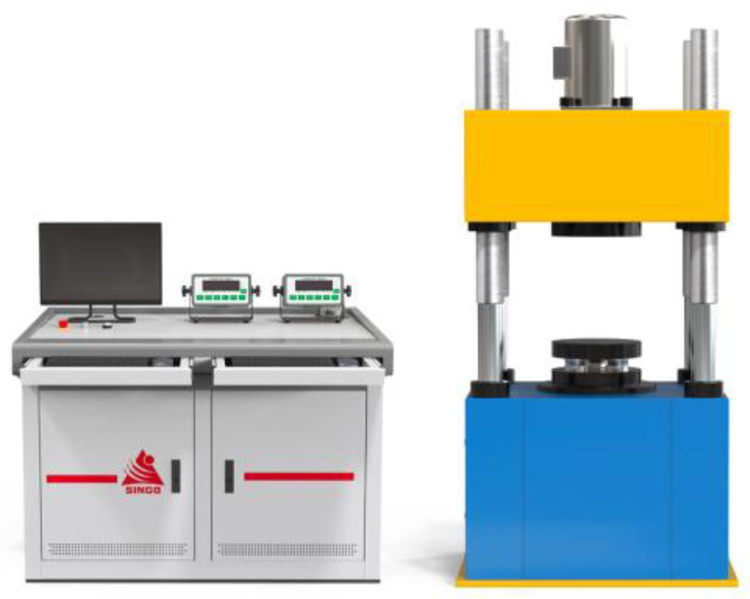
Superimposed force standard device.

**Figure 2 micromachines-16-01069-f002:**
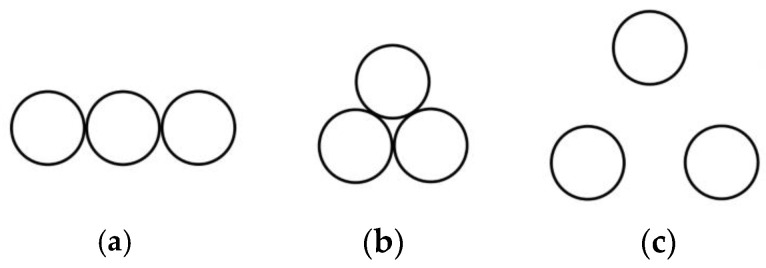
Three sensor placement modes. (**a**) Single font; (**b**) Tight circle; (**c**) “Pin font”.

**Figure 3 micromachines-16-01069-f003:**
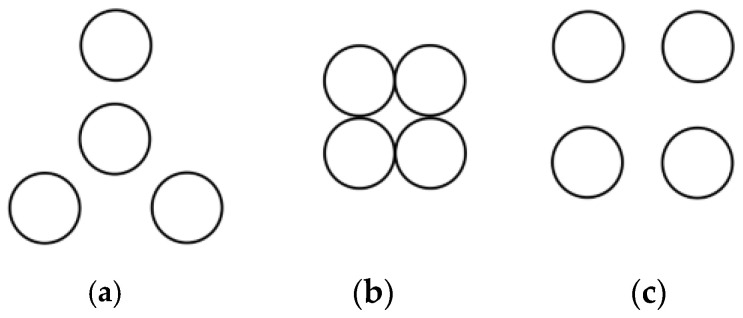
Four sensor placement modes. (**a**) “3 + 1” placement; (**b**) Tight square; (**c**) Sparse square.

**Figure 4 micromachines-16-01069-f004:**
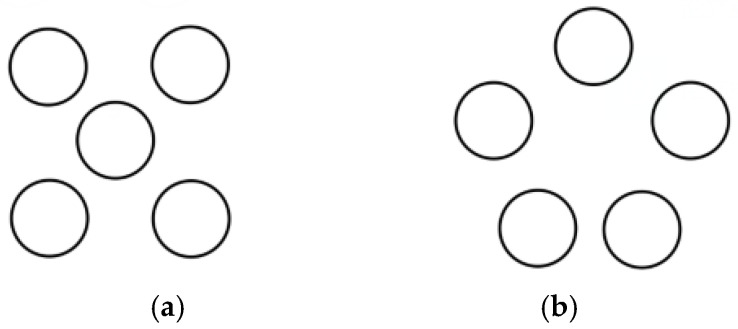
Five sensor placement modes. (**a**) “4 + 1” placement; (**b**) Sparse ring.

**Figure 5 micromachines-16-01069-f005:**
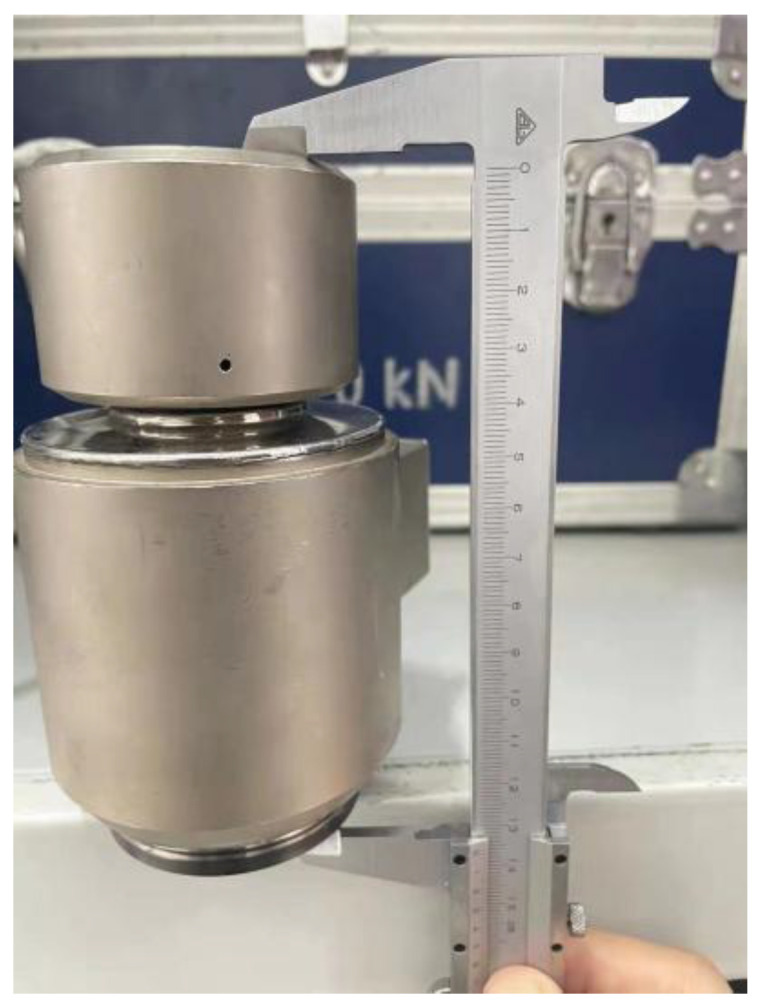
Sensor height.

**Figure 6 micromachines-16-01069-f006:**
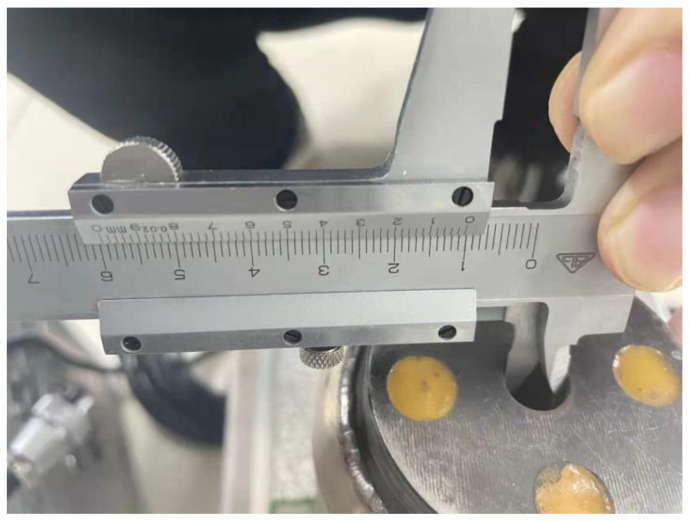
The diameter of the threaded hole.

**Figure 7 micromachines-16-01069-f007:**
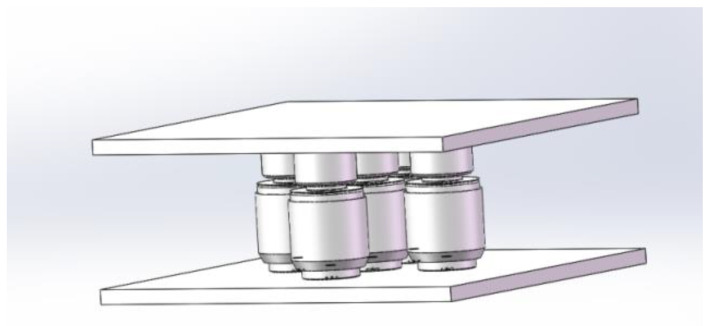
Model of superimposed force measurement instrument.

**Figure 8 micromachines-16-01069-f008:**
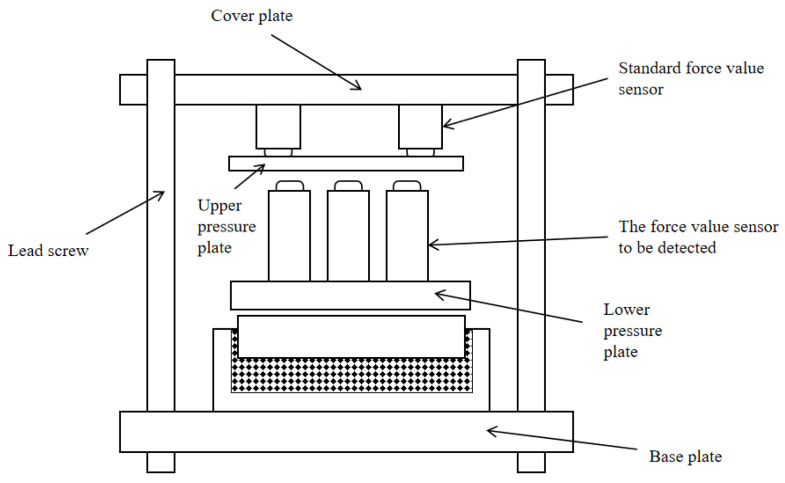
Schematic diagram of the superimposed force value testing machine.

**Figure 9 micromachines-16-01069-f009:**
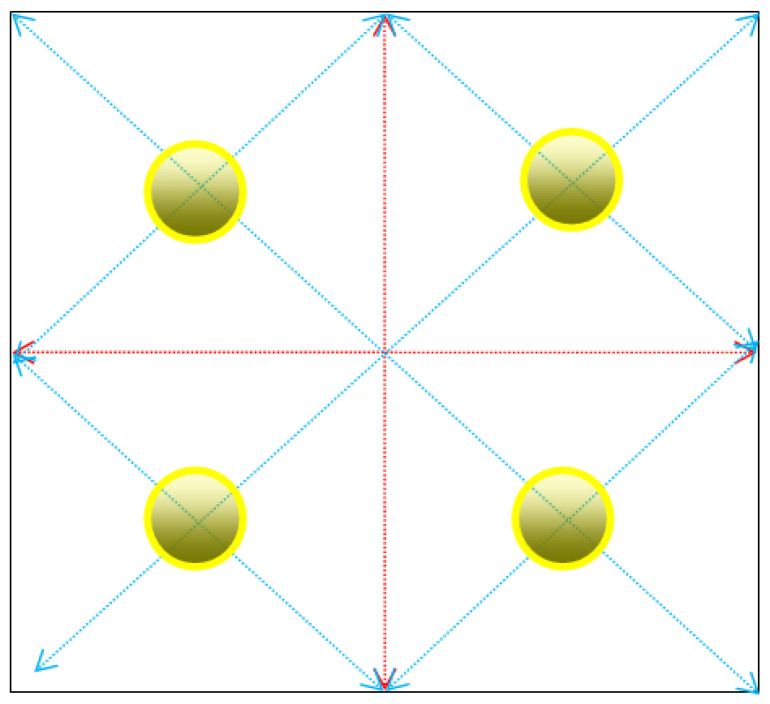
Schematic diagram of the force application point of the upper pressure-bearing plate.

**Figure 10 micromachines-16-01069-f010:**
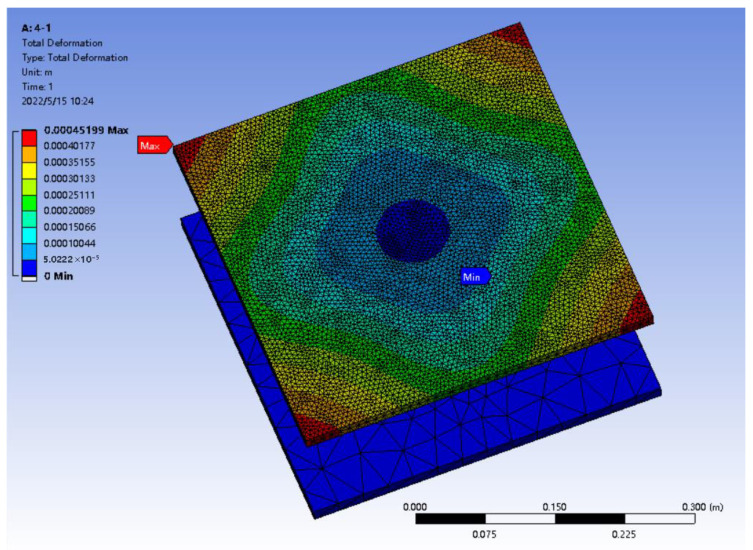
Total deformation amount.

**Figure 11 micromachines-16-01069-f011:**
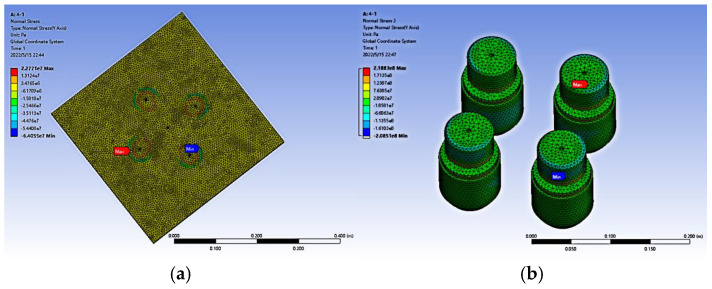
Stress diagram. (**a**) Cover stress diagram; (**b**) Sensor stress diagram.

**Figure 12 micromachines-16-01069-f012:**
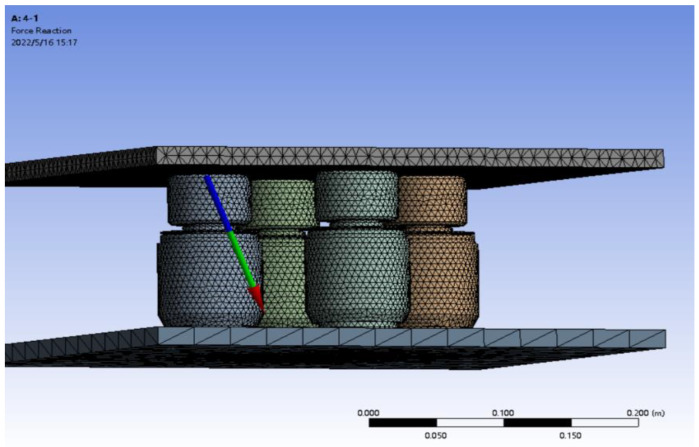
The expression method of contact surface force.

**Figure 13 micromachines-16-01069-f013:**
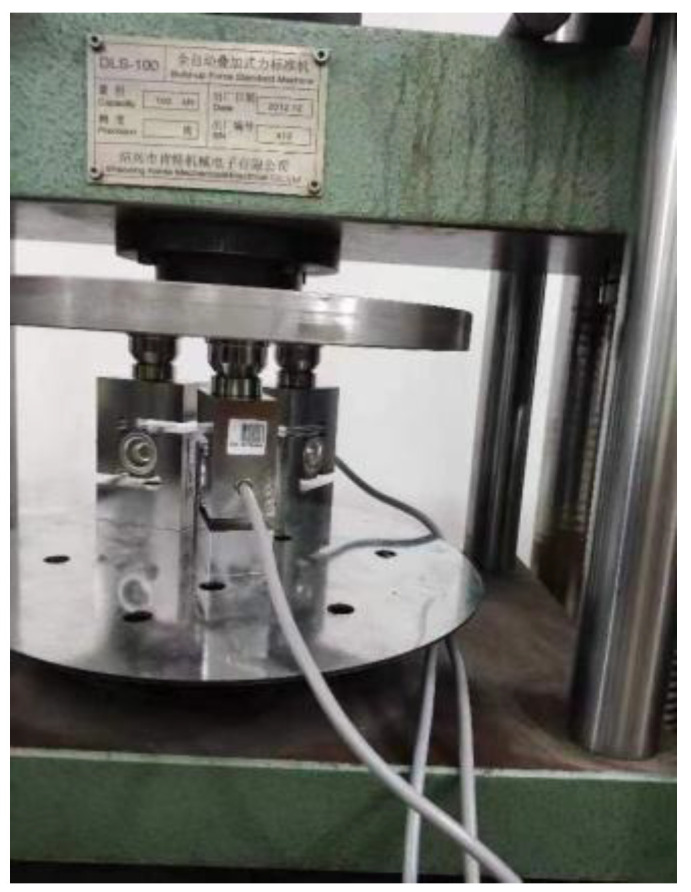
Test device.

**Table 1 micromachines-16-01069-t001:** Structural steel attribute parameters.

Materials	Density	Tensile Yield Strength	Compressive Yield Strength	Tensile Strength Limit
Structural steel	7850 kg × m^3^	250 MPa	250 MPa	460 MPa

**Table 2 micromachines-16-01069-t002:** Deformation of different models.

Number of Sensors	Placement Method	Maximum Deformation	Location
3	Single font	/	/
3	Tight circle	0.53148 mm	Top corner
3	“Pin font”	0.42848 mm	Top corner
4	Tight square	0.62565 mm	Top corner
4	Sparse square	0.45199 mm	Top corner
4	“3 + 1”	0.54681 mm	Top corner
5	“4 + 1”	0.57020 mm	Top corner
5	Sparse ring	0.60385 mm	Top corner

**Table 3 micromachines-16-01069-t003:** Maximum stress of different models.

Number of Sensors	Placement Method	Upper Bearing Plate	Maximum Stress	Upper End Face of the Sensor	Maximum Stress
3	Single font	/	/	/	/
3	Tight circle	Contact surface	21.450 MPa	Contact surface	21.451 MPa
3	“Pin font”	Contact surface	20.358 MPa	Contact surface	20.352 MPa
4	Tight square	Contact surface	33.891 MPa	Contact surface	33.281 MPa
4	Sparse square	Contact surface	22.771 MPa	Contact surface	22.765 MPa
4	“3 + 1”	Contact surface	25.835 MPa	Contact surface	25.835 MPa
5	“4 + 1”	Contact surface	29.744 MPa	Contact surface	29.744 MPa
5	Sparse ring	Contact surface	32.513 MPa	Contact surface	32.548 MPa

**Table 4 micromachines-16-01069-t004:** Three sensor force values.

Arrangement Mode	The First Component	Second Component	The Third Component	Force Sum	Error
Single font	/	/	/	/	/
Tight circle	10,008.2 N	9993.1 N	9998.9 N	30,000.2 N	6.6678 × 10^−6^
“Pin font”	10,004.3 N	9997.3 N	9998.5 N	30,000.1 N	3.333 × 10^−6^

**Table 5 micromachines-16-01069-t005:** Four sensor force values.

Arrangement Mode	The First Component	Second Component	The Third Component	The Fourth Component	Force Sum	Error
Tight square	10,015 N	9986.5 N	10,008 N	9992.3 N	40,001.8 N	4.5 × 10^−5^
Sparse square	10,010 N	9989.8 N	10,012 N	9988.4 N	40,000.2 N	5.0 × 10^−6^
“3 + 1” placement	10,000.5 N	10,004.6 N	9994.2 N	10,000.1 N	39,999.4 N	1.5 × 10^−5^

**Table 6 micromachines-16-01069-t006:** Five sensor force values.

Arrangement Mode	The First	Second	The Third	The Fourth	The Fifth	Force Sum	Error
“4 + 1”	10,000.8 N	9988.4 N	10,012 N	9994.2 N	10,006.7 N	50,002.1 N	4.2 × 10^−5^
Sparse ring	10,005.2 N	10,003.7 N	9995.6 N	10,001.9 N	9992.6 N	49,999.0 N	2.0 × 10^−5^

**Table 7 micromachines-16-01069-t007:** Sensor indication.

Force Standard Machine (kN)	31715918 (kN)	52043494 (kN)	52043493 (kN)	Actual Parallel (kN)	Theoretical Parallel (kN)	Error
3	0.9690	1.0548	0.9768	3.0006	/	/
6	1.9445	2.0772	1.9776	5.9993	5.9959	0.06%
9	2.9473	3.0960	2.9575	9.0008	8.9957	0.06%
12	3.9590	4.1610	3.8773	11.9973	11.9938	0.03%
15	4.9705	5.2264	4.7997	14.9966	14.9922	0.03%
18	5.9808	6.2914	5.7241	17.9963	17.9912	0.03%
21	6.9859	7.3634	6.6473	20.9966	21.001	−0.02%
24	7.9894	8.4310	7.5758	23.9962	24.0114	−0.06%
27	8.9929	9.4980	8.5051	26.996	27.0038	−0.03%
30	9.9959	10.5647	9.4364	29.997	29.9966	0.00%

## Data Availability

The original contributions presented in this study are included in the article.

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
