# Peer review of "Design and Research of Superimposed Force Sensor"

_micromachines, 2025, doi:10.3390/mi16091069_

Round 1

Reviewer 1 Report

Comments and Suggestions for Authors

Thank you, the work is very interesting and engaging. As always, an outside reader has many questions that do not always meet the expectations of a researcher, but a slightly improved version of your manuscript will certainly be of great benefit to researchers in this field. Thank you, it was very interesting to read.

Author Response

Thank you very much for pointing out the shortcomings of this article. I have revised the entire manuscript based on your comments. The relevant content is as follows:

Comments 1: Keywords: not necessary, but perhaps keywords can be added SolidWorks, FEM, ANSYS for searching comfortability, because this work can be useful for beginners.

Respones 1: Thank you for pointing this out. I agree with this comment. Therefore, I have added the keywords solidworks and ansys. It can be found on line 28 of the manuscript.

Comments 2: Introduction: can be expanded with a statement about why this research is important compared to previous works of other researchers.

Respones 2: Thank you for pointing this out. I agree with this comment. Therefore, I have already explained in the introduction why this research is more important than the previous work of other researchers. It can be found on line 28 of the manuscript. It can be found on lines 72 to 79 and 86 to 91 of the article.

Comments 3: Materials and Methods: The title of Chapter 2 begins with the word Materials, but the material itself is only mentioned in the last part of this chapter.  Perhaps it would be worth moving it to the beginning of the chapter to maintain the logic of the title?

Respones 3: Thank you for pointing this out. I agree with this comment. There is a lack of logic between the title and the text content. Therefore, I modified the title to better summarize what this section does and make the logic more distinct. It can be found on line 92 of the article.

Comments 4: In figure 1, perhaps you can mark the relevant details, because someone who sees this device for the first time and is interested in it, may get the wrong conclusions about the equipment. The reader be interested in what is hidden behind the door marked with a red line. Perhaps this image could be improved with a diagram or something that would give more information about the equipment and working principles.

Respones 4: Thank you for pointing this out. I agree with this comment. I have roughly introduced the components and functions of the device so that readers can have a clear understanding of it.  It can be found on lines 98 to 104 of the article.

Comments 5: 2.3. maybe you could define what actions are called pre-processing, what to expect in this subsection. I am curious: How the experiment setup enables to avoid shear force or it is not working in this device? It might be useful to mark where the transducer is in the picture. It remains unclear what the purpose of this section is. How preparation is related to finite element simulation. Figure 8 name please all the parts drawn in the picture. Sometimes the reader is not clear about the material or the function of the part.

Respones 5: Thank you for pointing this out. I agree with this comment. I introduced the purpose of this chapter and the issues to be noted in the pre-processing of simulation. I have made comprehensive annotations of each part of the structure in Figure 8 to facilitate readers' understanding of the schematic diagram. It can be found on lines 164 to 170 of the article.

Comments 6: 2.4. Is mentioned structural steel, line 177. Next to the first mention of the material, it would be good to name the specific type of steel used, the manufacturer from which it was purchased, and the technical conditions of this product. Have you tried experimenting with other steels or metal alloys? If not, maybe it would be worth doing some research with other steel samples? This is not a requirement, just very interesting. The numerical values in the table 2 provided are given without specifying the errors. Did you derive the average from the measurements or in some other way obtain the specific numbers without the error values? If this does not contradict your research concept, perhaps it could be supplemented with error values?

Respones 6: Thank you for pointing this out. I agree with this comment. For the material Settings of the model in ANSYS, I set the materials of the simulation model based on the main materials of the test equipment. We only know the material composition of the test equipment, but we are not very familiar with the manufacturers who purchased the steel and the technical conditions of the product. If there is an opportunity, we will also conduct experiments using other steels or metal alloys. The values in Table 2 are specific figures obtained through simulation. It can be found on lines 196 to 201 of the article.

Comments 7: The text mentions finite element analysis, it would be interesting to see more data in your work, it would enrich your work if you agree.

Respones 7: Thank you for pointing this out. I agree with this comment. If possible later, I will try some other simulation processes.

Comments 8: 3.1. In Table 2, the columns location, minimum deformation, location 2, do not make sense to print due to their fixed size. Perhaps these columns could be removed from the table and these parameters mentioned in the table description. How table 2 results compare with figure 10? How you measured 0.42848mm with such accuracy? Please name the details.

Respones 8: Thank you for pointing this out. I agree with this comment. In Table 2, I have removed the minimum deformation and its position, and described this phenomenon in the text. 0.42848mm is the deformation of the cover plate obtained through simulation under an applied force of 10,000N × the number of sensors. It can be found on lines 243 to 254 of the article.

Comments 9: 3.2. Table 3 needs the same correction with removing constant values. Describe FE regions in fig. 11 in the text. May be some domain info, virtual power or something you got from FEM.

Respones 9: Thank you for pointing this out. I agree with this comment. I have made relevant descriptions of Figure 11 in the text. It can be found on lines 279 to 283 of the article.

Comments 10: 3.3. It is not clear the arrow purpose in figure 12. a more detailed discussion of the figures of simulations are needed

Respones 10: Thank you for pointing this out. I agree with this comment. I have described the display information and related uses of the arrows in the text. It can be found on lines 308 to 316 of the article.

Comments 11: 4. Discussion. May be text part from line283 to288 and figure 13 is more useful in the part 2?

Respones 11: Thank you for pointing this out. I have made corrections to the title of the second part, so the content of this part may no longer be suitable for the relevant expressions in the second part.

Reviewer 2 Report

Comments and Suggestions for Authors

To improve the measurement accuracy and equipment stability of superposition-type force sensors, the authors analyzed the influence of the number of paralleled individual sensors and their layout within a superposition-type force measurement instrument, and suggested the three single force sensors in a “Pin font” shape. This is hard and heavy working definitely. However, I have queries about the work.

1. The title of the paper is "Design and research of superimposed force sensor", but it does not explicitly mention in Sec. 2.1. Instead, it introduces the design scheme of a superposition-type force standard device, why?

2. where is the superimposed-type force sensors in Fig.1?

3. Fig.6 shows a threaded hole. Where is the hole drilled, in sensor or upper platen?

4. In Sec.2.2, a 0.2mm recess was added to the contact surface between the upper side of the lower platen and the sensor. Where is the recess in Fig.7? and why can the recess increase the contact area and enhance the overall stability?

5. Fig.8 should be labeled more clear. And the text “the upper cover plate” in the context and “Upper pressure plate” in Fig.8 refer to the same plate?

6. In sec.2.4, “the applied force is symmetrical throughout the …”, how can the force be symmetrically applied?  It is not explained clearly in Fig.9.

7.In Sec.3.1, how did the total deformation amount,Fig.10, be derived without a force being applied on the upper plate?

8. Was the material of the sensors assumed to be structural steel In Sec.3.2? Will it affect the overall analysis?

9. What is the meaning of the number, “31715918”,”52043494”, ”52043493”, in Table.7?

Comments on the Quality of English Language

The quality of English Language seems quite good.

Author Response

Thank you very much for pointing out the shortcomings of this article. I have revised the entire manuscript based on your comments. The relevant content is as follows:

Comments 1: The title of the paper is "Design and research of superimposed force sensor", but it does not explicitly mention in Sec. 2.1. Instead, it introduces the design scheme of a superposition-type force standard device, why?

Respones 1: Thank you for pointing this out. In order to design a superimposed force sensor with excellent performance, we first designed multiple superimposed schemes in Section 2.1. Later, we analyzed the working characteristics of these schemes respectively to identify the more outstanding one. Finally, we verified the design scheme through experiments.

Comments 2: where is the superimposed-type force sensors in Fig.1?

Respones 2: Thank you for pointing this out. In Figure 1, the left part is the microcomputer control system, and the right part is the superimposed standard force gauge. I have modified the text content to enable readers to better understand the figure. It can be found on lines 98 to 104 of the article.

Comments 3: Fig.6 shows a threaded hole. Where is the hole drilled, in sensor or upper platen?

Respones 3: Thank you for pointing this out. The threaded holes are drilled on the pressure-bearing plate. Drilling threaded holes on the pressure-bearing plate is to better cooperate with the sensor placed in the middle. Figure 6 shows the measurement process of the diameter of the sensor pressure head. The diameter of the threaded hole is the same as that of the sensor pressure head. It can be found on lines 140 to 148 of the article.

Comments 4: In Sec.2.2, a 0.2mm recess was added to the contact surface between the upper side of the lower platen and the sensor. Where is the recess in Fig.7? and why can the recess increase the contact area and enhance the overall stability?

Respones 4: Thank you for pointing this out. When modeling, a 0.2mm groove is added at the contact position between the lower pressure-bearing plate and the sensor. The sensor can be embedded in this groove, so that both the front and side of the sensor can come into contact with the pressure-bearing plate, avoiding the displacement of the sensor's position during the force application process and enhancing its stability.

Comments 5: Fig.8 should be labeled more clear. And the text “the upper cover plate” in the context and “Upper pressure plate” in Fig.8 refer to the same plate?

Respones 5: Thank you for pointing this out. The "upper cover plate" in the text is the same plate as the "upper pressure plate" in Figure 8. I have made detailed annotations on Figure 8 so that readers can understand this schematic diagram more clearly. At the same time, I have also corrected the wording to make it more in line with the annotations in the figure.

Comments 6: In sec.2.4, “the applied force is symmetrical throughout the …”, how can the force be symmetrically applied?  It is not explained clearly in Fig.9.

Respones 6: Thank you for pointing this out. The four positions for applying force should be at equal distances and symmetrically distributed from the center and edge of the upper pressure plate. The center positions of the four sides of the upper pressure plate can be marked to divide the upper pressure plate into four identical areas. A diagonal can be taken for each area, and the intersection point of the diagonals is a position for applying force. It can be found on lines 209 to 214 of the article.

Comments 7: In Sec.3.1, how did the total deformation amount,Fig.10, be derived without a force being applied on the upper plate?

Respones 7: Thank you for pointing this out. The upper pressure-bearing plate deforms under an external force of 10,000 N× the number of sensors. I have also supplemented the deformation rules and states in the manuscript. It can be found on lines 243 to 254 of the article.

Comments 8: Was the material of the sensors assumed to be structural steel In Sec.3.2? Will it affect the overall analysis?

Respones 8: Thank you for pointing this out. During the simulation process, the material of the sensor model was set based on the main material composition of the physical object. Therefore, I supplemented the definition of the sensor material. It can be found on lines 196 to 201 of the article.

Comments 9: What is the meaning of the number, “31715918”,”52043494”, ”52043493”, in Table.7?

Respones 9: Thank you for pointing this out. To facilitate the distinction of the force values measured by each sensor, they are numbered as 31715918, 52043494, and 52043493 based on the production information of the sensors. This number can also be defined as something else.

Reviewer 3 Report

Comments and Suggestions for Authors

The manuscript entitled “Design and Research of Superimposed Force Sensor” reports on the design, finite element analysis (FEA), and partial experimental validation of a superimposed force standard device. The topic is relevant to the readership of Micromachines, as force standard machines play a key role in metrology, calibration, and sensor technology. The use of SolidWorks modeling and ANSYS simulation, combined with experimental comparison, provides useful insights. However, the manuscript in its current form suffers from organizational issues, insufficient depth in scientific discussion, and unclear presentation of figures and results. Major revisions are required before the manuscript can be considered for publication.

  1. Figures are low in resolution and captions are minimal. It is difficult for readers to follow which results support which claims. The author should follow the format requirements of the journal.
  2. Group related figures to better support the logical flow of the text.
  3. While FEA results are detailed (deformation, stress, force distribution), the experimental section is limited to a single setup with three sensors
  4. Reported errors are in the order of 10⁻⁴ to 10⁻³ %, which seems overly optimistic and may raise credibility concerns. No discussion is provided on measurement uncertainties, error sources, or limits of FEA modeling.
  5. The statistical data and error distribution of multiple measurements should be increased to improve the credibility of the data.
  6. References are heavily skewed toward Chinese sources, some of which may not be easily accessible to the international readership. Some links in the reference are inaccessible. Including more international works would strengthen the background.

Author Response

Thank you very much for pointing out the shortcomings of this article. I have revised the entire manuscript based on your comments. The relevant content is as follows:

Comments 1:Figures are low in resolution and captions are minimal. It is difficult for readers to follow which results support which claims. The author should follow the format requirements of the journal.

Respones 1: Thank you for pointing this out. I agree with this comment. I supplemented the explanatory text for each picture and table. It can be found on lines 98 to 104, 140 to 148, 164 to 170, 210 to 215, 243 to 254, 279 to 283 and 308 to 316 of the article.

Comments 2: Group related figures to better support the logical flow of the text.

Respones 2: Thank you for pointing this out. I agree with this comment. I have adjusted and grouped the pictures that do not conform to the text logic so that readers can better understand the article.

Comments 3: While FEA results are detailed (deformation, stress, force distribution), the experimental section is limited to a single setup with three sensors

Respones 3: Thank you for pointing this out. I agree with this comment. Due to our limited resources, we can only first obtain a relatively ideal scheme through simulation and then verify it through experiments. If there is an opportunity in the future, we will try to verify other solutions.

Comments 4: Reported errors are in the order of 10⁻⁴ to 10⁻³ %, which seems overly optimistic and may raise credibility concerns. No discussion is provided on measurement uncertainties, error sources, or limits of FEA modeling.

Respones 4: Thank you for pointing this out. I agree with this comment. The magnitude of this error is the deviation of the Y-direction force value obtained through simulation, which may be smaller than the actual one. Therefore, we conducted an analysis through experiments. The experimental error is greater than the simulation error, which may be related to measurement uncertainty, the limitations of finite element modeling, and other sources of error.

Comments 5: The statistical data and error distribution of multiple measurements should be increased to improve the credibility of the data.

Respones 5: Thank you for pointing this out. I agree with this comment. We conducted a further analysis of the data and errors obtained from the experiment, summarized the connections and differences between theory and experiment, and enhanced the credibility of the data. It can be found on lines 367 to 372 of the article.

Comments 6: References are heavily skewed toward Chinese sources, some of which may not be easily accessible to the international readership. Some links in the reference are inaccessible. Including more international works would strengthen the background.

Respones 6: Thank you for pointing this out. I agree with this comment. I read some relevant foreign language literature and added and replaced some Chinese literature. It can be found on lines 416, 420, 430 and 475 of the article.

Round 2

Reviewer 3 Report

Comments and Suggestions for Authors

The author has addressed all my concerns